# Motor Dynamic Loading and Comprehensive Test System Based on FPGA and MCU

Chunxiang Zhu [1], Linxin Bao [1], Bowen Zheng [2], Jiacheng Qian [1], Yongdong Cai [1] and Binrui Wang [3,*]

1   Engineering Training Center, China Jiliang University, Hangzhou 310018, China;
    16a2900069@cjlu.edu.cn (C.Z.); 1900201509@cjlu.edu.cn (L.B.); 2000303206@cjlu.edu.cn (J.Q.);
    1800304230@cjlu.edu.cn (Y.C.)
2   School of Electronic Information, Hangzhou Dianzi University, Hangzhou 310018, China;
    bwzheng@hdu.edu.cn
3   School of Mechanical and Electrical Engineering, Hangzhou Dianzi University, Hangzhou 310018, China
*   Correspondence: wangbrpaper@163.com

**Abstract:** In view of the problem that the traditional motor test system cannot directly test the transient parameters of the motor and the dynamic arbitrary load loading requirements during motor loading, as well as the high cost of implementation, this research uses STM32+FPGA as the core to form the main control of the motor test system unit, combining the superior control performance of the ARM processor and the high-speed data processing advantages of FPGA. FPGA and STM32 are controlled by the FSMC bus communication and data ping-pong algorithm. Using this method, a small-size control core board in the motor test system is manufactured. It can be embedded in the existing traditional dynamometer system to improve the dynamometer transient parameter test and the dynamic motor loading performance. The experimental results show that the system can basically meet the requirements of the motor transient test and dynamic loading, and can achieve the fastest data refresh rate of 1 ms when measuring the motor's speed and torque, as well as arbitrary waveform loading within a 100 M sampling frequency, with a loading error of 0.8%. It satisfies the motor transient test and dynamic loading requirements.

**Keywords:** motor transient test; dynamic loading; STM32+FPGA; FSMC; ping-pong algorithm

## 1. Introduction

In recent years, the application scope of motors has been expanding, and the universality of motor operating conditions determines the complexity of the motor experiment. Traditional testing devices and methods are time-consuming, use a wide range of instruments, and have a low precision, resulting in low intelligence and low efficiency, which greatly affect the accuracy and quality of testing. Most motor tests use a dynamometer as a loading instrument, connected to the motor being tested, and then simultaneously interpret different motors as required. The dynamometer is controlled by the central control computer, and then different test items are analyzed by the power analyzer. A dynamometer is one of the important devices for performance and equipment testing in motor manufacturing and product R&D. It can be used to simulate and control the load of the tested motor so as to measure the torque, speed, current, voltage, power, efficiency, and other parameters of the motor, as well as for other special dynamic test items, such as safety test, dynamic balance test, NVH test (vibration/noise/life), etc. [1–6]. With the development of modern technology, in addition to conventional items such as motor speed, torque, voltage, and current, the requirements for the motor transient performance test are also increased [7–10]. In the traditional dynamometer system, the loads can only be loaded one by one, but the motor performance parameters are measured for a certain load point, which cannot meet the transient performance test of the motor. The loading accuracy and testing accuracy cannot meet the modern requirements. Therefore, enterprises have put forward

the demand for high-performance loading and testing equipment. The equipment needs to meet the high bandwidth and high-precision loading and measurement of a variety of different waveforms, including step signal, square wave, trapezoidal wave, and sine wave. The measurement error should not be higher than 0.5%, and the loading error should not be higher than 0.2% [11–13]. According to the above analysis, the key technical points of the transient performance test of the motor and driver mainly include the following two aspects: First, the loading function of any load curve, that is, it can provide step, sine wave, square wave, sawtooth wave, and even arbitrary waveform loading of torque or speed [14–17]. The second is parameter transient waveform measurement, including the parameter transient waveform measurement of torque, speed, voltage, current, efficiency, and speed torque curve. According to this technical point, in order to meet the high-speed loading and high-speed processing at the same time, this paper innovatively designs a dual core dynamometer loading and testing integrated control system with FPGA and MCU communicating through FSMC. Regarding the industrial application requirements of the system, the corresponding software and hardware functions are designed to meet the requirements of various waveform loadings and measurements of different motor platforms, as well as the requirements of high-speed communication and responses with industrial computers. Therefore, this paper implements a motor dynamic loading and transient parameter testing device. In the process of motor testing, by loading any load curve and testing the transient performance of the motor, it can meet the use requirements of dynamometer equipment embedded in motor testing enterprises [18–20].

## 2. Composition Principle of the Motor Test System

High-speed measurements and dynamic loadings of the motor transient parameters are realized, as shown in the structural block diagram shown in Figure 1. The dynamometer is coaxially connected to the tested motor. Various loads are loaded on the tested motor through the dynamometer control system so as to simulate the torque, speed, current, voltage, power, efficiency, and other parameters of the motor in actual operation, as well as other special dynamic test items, such as the safety test, dynamic balance test, and NVH (vibration and noise life) test. When loading, the upper computer sends instructions to control the free loading engine to output the control signal, which is input to the dynamometer drive controller. After the power of the controller is amplified, the control signal is loaded on the load motor so as to drive the load motor in order to load the tested motor. At the same time, the torque−speed transducer feeds back the speed and torque of the measured motor to the free loading engine. The free loading engine adjusts the output of the load in real time according to the feedback value to realize constant speed loading or constant torque loading. The torque calculation formula is as follows:

$$T_P = N(f - f_0)/(f_P - f_0) \tag{1}$$

$$T_r = N(f_0 - f)/(f_0 - f_r) \tag{2}$$

where $T_p$ is the forward torque, $T_r$ is the reverse torque, $N$ is the torque full scale, $f_0$ is the torque zero output frequency value (KHz), $f_p$ is the forward full-scale output frequency value (KHz), $f_r$ is the reverse full scale output frequency value (KHz), and $f$ is the measured torque output frequency value (KHz). The speed calculation formula is as follows:

$$S = 60 \times f/Z \tag{3}$$

where $S$ is the rotational speed (rpm), $f$ is the measured rotational speed output frequency (Hz), and $Z$ is the number of teeth (lines) of the speed-measuring code disk of the sensor.

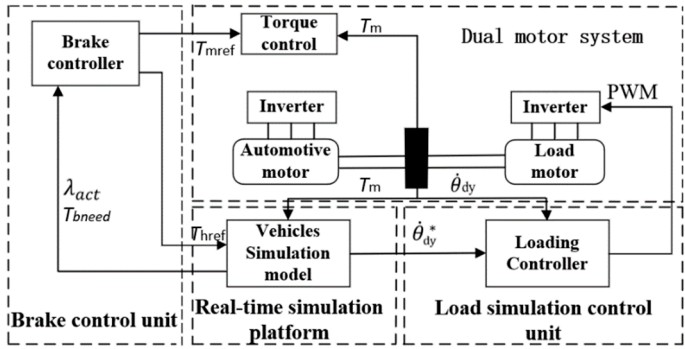

**Figure 1.** System structure block diagram.

## 3. System Hardware Design

### 3.1. Overall Structure

The circuit structure of the motor test system based on STM32 and FPGA is shown in Figure 2. The system is composed of a user-defined signal output and a speed and torque acquisition circuit. Firstly, STM32 receives the control signals, such as the loading waveform type and waveform parameters, sent by the PC ethernet port, which generates the corresponding analog output or digital output, and obtains the output control signal through FMSC communication, FPGA analysis, magnetic coupling isolation circuit, and then the conversion of the corresponding functional modules. This signal is input to the tested motor or load motor driver to drive the corresponding motor. Secondly, when the real-time data waveform needs to be uploaded, STM32 receives the acquisition data command sent by the PC ethernet. After the speed torque feedback signal is sampled by the magnetic coupling isolation, level conversion circuit, and ADC, it is stored in the on-chip dual port RAM of FPGA, and then STM32 reads the data in the way of a ping-pong operation through the FMSC interface and sends it to the upper computer through the ethernet. The composition and functions of each part of the circuit are introduced below. The components are described below.

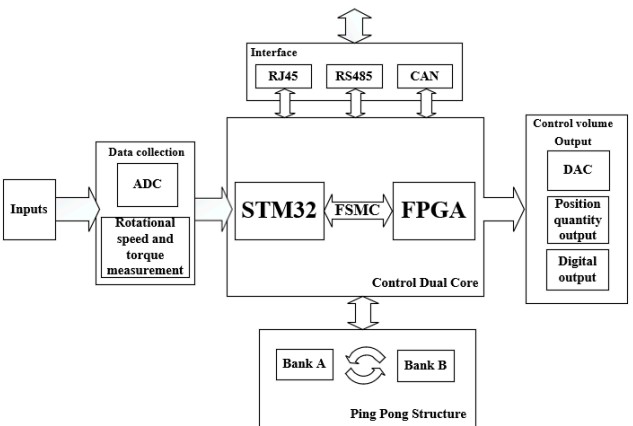

**Figure 2.** Hardware block diagram.

### 3.2. Circuit Design of STM32 Module

The minimum system circuit structure of the STM32 module is shown in Figure 3. The STM32 system selects STM32H743ZI2, with ARM-Cortex M7 as the core as the main control chip. The STM32H743ZI2 chip is a 32-bit microprocessor. It not only has rich peripheral resources, but also has a fast data processing speed and low power consumption. It is very suitable for occasions with high data speed and high system response requirements. According to the functional requirements of the project, circuits including the ethernet communication circuit, digital output circuit (DO), low-speed ADC circuit, display key circuit, SDRAM data cache, and so on, are designed. The ethernet communication circuit is

mainly responsible for the data transmission between the upper computer and the lower computer. The digital output circuit mainly realizes the digital output control relay to pull in or open. The low speed ADC is mainly used to collect the working environment, such as the external temperature/humidity. The display key circuit realizes the human−computer interaction function. The SDRAM memory circuit mainly realizes the high-speed data cache.

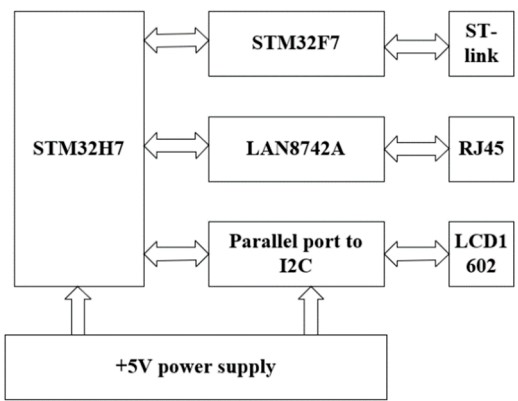

**Figure 3.** STM32 module minimum system circuit block diagram.

### 3.3. FPGA Module Circuit Design

The main control chip used in FPGA is the fourth generation EP4CE40F23C8N, which has rich logic resources and about 40,000 gates. There are about 532 user IO ports, so it is very suitable for multi-channel data acquisition and synchronous processing. The FPGA main control core circuit mainly includes a 10M magnetic coupling isolation circuit, pulse output circuit (CO), speed/torque input circuit (CI), high-speed DAC analog output circuit, two-stage operational amplifier circuit, high-speed ADC analog acquisition circuit, low-pass filter circuit, and DDR3 data buffer circuit. It mainly completes the data ping-pong operation, analog quantity acquisition and output, speed and torque measurements, receiving control commands, outputting pulse signals, receiving encoder signals, and other functions of motor testing.

The structural diagram is shown in Figure 4.

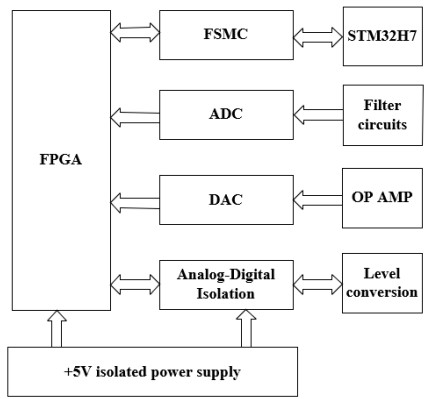

**Figure 4.** FPGA core circuit block diagram.

### 3.4. FSMC Communication Design between STM32 and FPGA

FSMC, a flexible static storage controller, can be connected with synchronous or asynchronous memory and a 16-bit PC memory card. The FMSC interface of STM32H7 supports memory such as SRAM, NAND FLASH, NOR FLASH, and PSRAM. The communication between STM32H7 and FPGA adopts a parallel FSMC bus. The collected data are saved by constructing a dual port RAM built in FPGA, and STM32H7 accesses it through the FMSC bus. According to the requirements of the collected data volume, the FMSC bus

adopts a 24-bit address line and 16-bit data line, and contains six control signals at the same time. The STM32 chip selects FPGA to read and write data. The structural block diagram is shown in Figure 5.

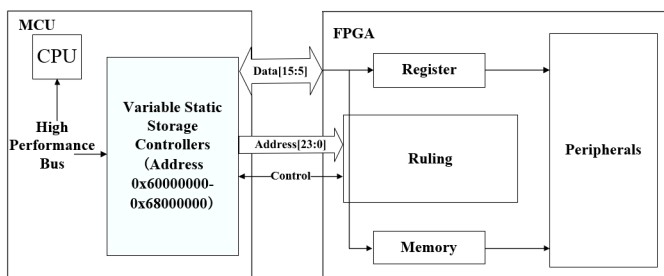

**Figure 5.** FMSC communication structure block diagram.

## 4. System Software Architecture

### 4.1. Overall Software Structure

The software design of the motor test system can be divided into the upper computer and lower computer. As shown in Figure 6, the upper computer completes the functions of the man−machine interface, speed and torque loading, speed and torque transient data acquisition, control and display, ethernet connection between upper and lower computers, etc. The lower computer parses and executes the instructions sent by the upper computer, collects data, stores data, and uploads data in real time. The following section introduces the software design of each part seperately.

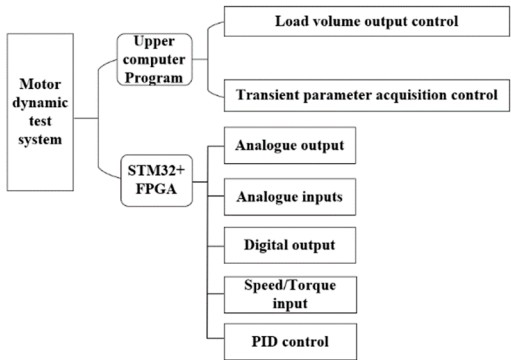

**Figure 6.** Software overall structure.

### 4.2. STM32 Software Design

The software architecture of the lower computer mainly includes the STM32H7 end and FPGA end. The main tasks of the STM32H7 terminal include analysis of the host computer instructions, real-time data transmission, TCP/IP connection, and low-speed ADC data acquisition. There are many tasks and complex functions, so it is necessary to reasonably plan the scheduling relationship between tasks. In order to improve the real-time performance of the software and reduce the coupling between tasks, the software is transplanted with the real-time embedded operating system—Ferrets. The operating system provides task management, time management, semaphore, message queue, memory management, and other functions, and the kernel supports both the priority scheduling algorithm and polling scheduling algorithm. The system is open source, tailorable, and free, which can effectively save the cost of instrument development and is suitable for industrial application. The software flow chart of the master operating system with the embedded real-time operating system FreeRTOS as the core is shown in Figure 7. After the system is initialized, the FreeRTOS operating system is initialized, and then the FPGA AO control thread is started at the same time so as to control the output of the analog quantity. The FPGA AI control thread is used to control the acquisition of the analog quantity. The

FPGA CI control thread is used to collect speed and torque. The FPGA PID control thread is used to control the speed and torque of the measured motor and load motor. The TCP server thread is used for data and instruction transmission, and has the function of TCP disconnection and reconnection.

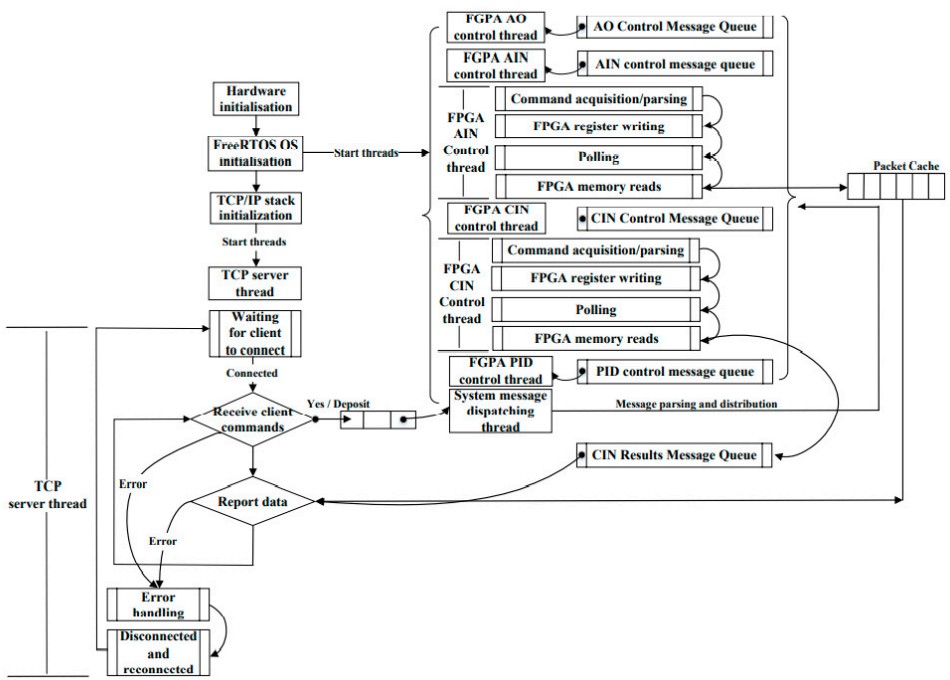

**Figure 7.** STM32H7 software flow chart.

### 4.3. FPGA Software Design

FPGA mainly completes external analog input acquisition, motor encoder signal input acquisition, data storage, receiving control instructions from ARM, controlling the analog output, controlling the position output, and controlling the digital output. The software architecture of FPGA is shown in Figure 8.

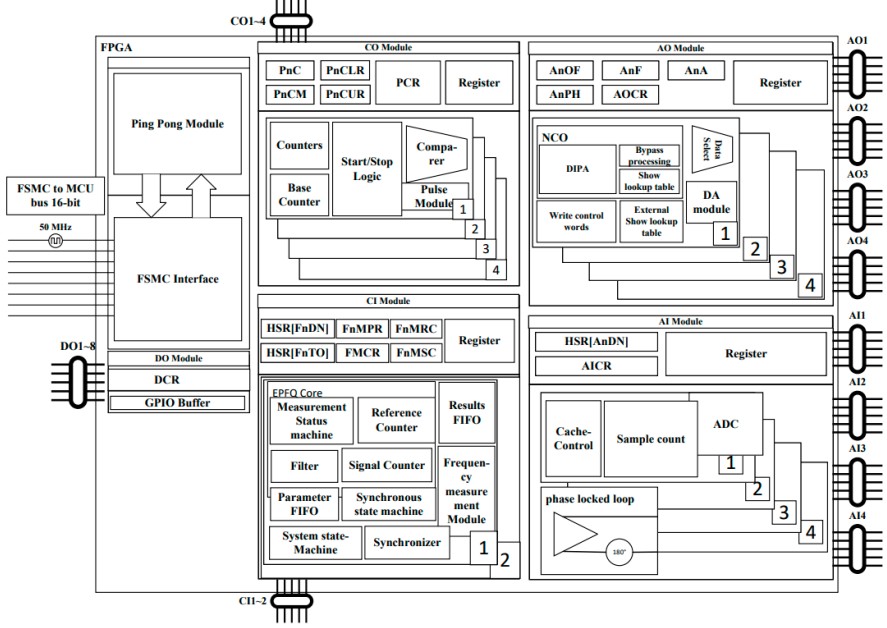

**Figure 8.** FPGA software flow chart.

*4.4. Ping-Pong Algorithm for Data*

Ping-pong operation is a common data flow control processing technique that can realize seamless transmission of data flow. The specific algorithm is shown in Algorithom 1 below. During the first buffering cycle, the input data are cached to Data Buffer Module 1. In the second buffering cycle, the input data are cached to Data Buffer Module 2, and the data in Data Buffer Module 1 are output to the data stream operation processing unit through the selection of "Output Data Stream Selection Unit". In the third buffering cycle, the data are cached into Data Buffer Module 1 by the input data stream selection unit, and Data Buffer Module 2 is selected by the output data stream selection unit to the data stream processing entity. In this design, the ADC data are driven by the ADC and enter the sampling module. After oversampling 200 times, the ADC data are sent to the data ping-pong module at a sampling rate of 1 KHz, so as to achieve a 1 ms data update rate. The ADC data acquisition process is shown in Figure 9.

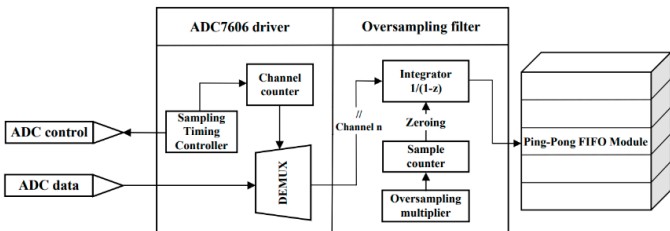

**Figure 9.** ADC data acquisition process.

---

**Algorithom 1** ping-pong operation

---

Input: DATA_Ti(i = 1,2,3, . . . *n*)
    if(buffer_1 ! = Full )
      buffer_1 = DATA_Ti
    else if (buffer_1 = Full)
      buffer_2 = DATA_Ti
    End if

---

## 5. Experimental Verification and Result Analysis

The system test mainly focuses on the measurement of the motor load and motor transient parameters. The test system is built with an SFT series electric dynamometer as the core, the model of the speed−torque sensor is the WSP speed−torque power sensor, and the output signals of the torque−speed are pulses with an amplitude of 5 V. The output range of the torque frequency signal is 5 KHz to 15 KHz, and the center frequency is 10 KHz. The output range of the speed frequency signal is 0 to 6 KHz, and the calculation formula for the speed and torque is given by Equations (1)–(3). The connection mode between the control system and the upper computer is 100 M ethernet. Figures 10 and 11 show the hardware platform of the project and PC control interface, respectively.

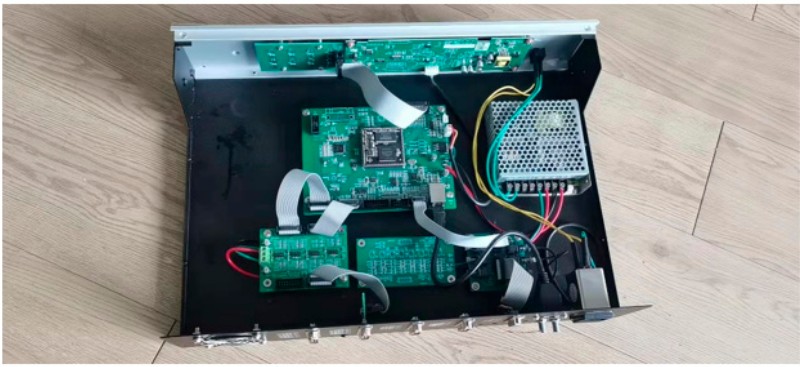

**Figure 10.** Motor test platform.

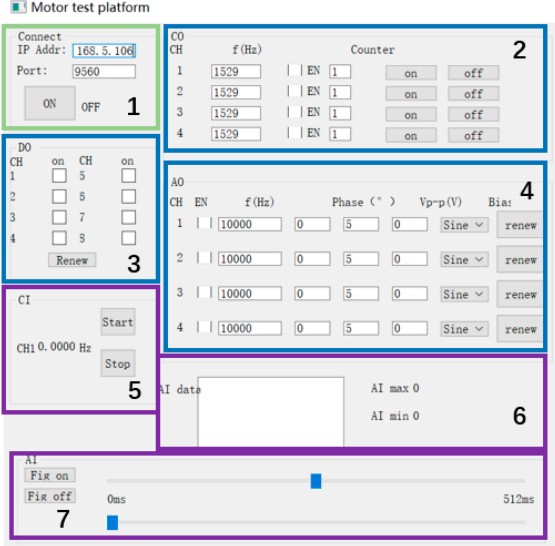

**Figure 11.** PC control interface.

### 5.1. Motor Load Test

The loading quantity includes the analog quantity output, pulse output, and digital quantity output. The analog quantity output also includes regular sine wave output, square wave output, triangular wave, and DC quantity output. The frequency range of the output of the three AC signals is 1 Hz to 10 MHz. The relative error of the frequency output is shown in Figure 12. It can be seen from the figure that the relative error of the frequency output is within 2%, and there is almost no error in the low frequency band.

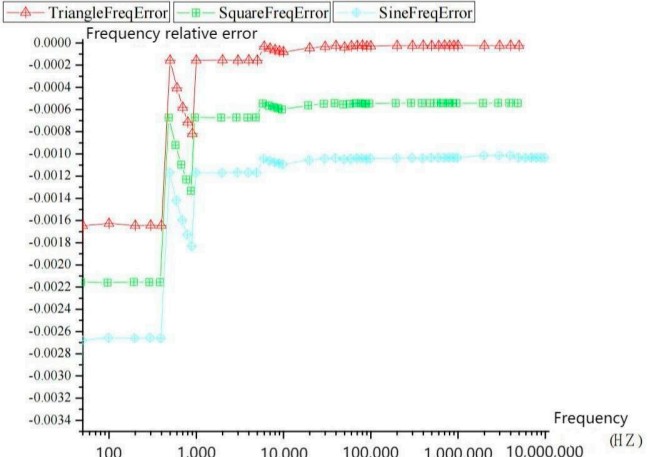

**Figure 12.** Relative error of the load output frequency.

The output range of the analog DC bias output voltage is −5 V to +5 V. The relative error of the DC bias output is shown in Figure 13. The abscissa represents the peak value and the ordinate represents the output relative error. As can be seen from the figure, the maximum relative error of the output peak value should not exceed 0.4% in order to meet the needs of practical application. The continuous pulse output mode can control the output frequency and phase, so as to control the forward and reverse rotation of the motor. The square wave loading amplitude error, sine wave loading amplitude error, and triangular wave loading amplitude error are shown in Figures 14–16, respectively. With the increase in frequency, the relative error increases slightly, and the relative error is within 0.8%.

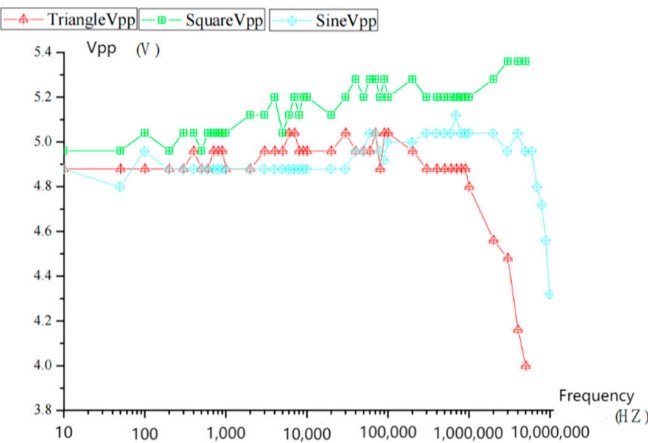

**Figure 13.** Real measurement when setting the 5 V bias output.

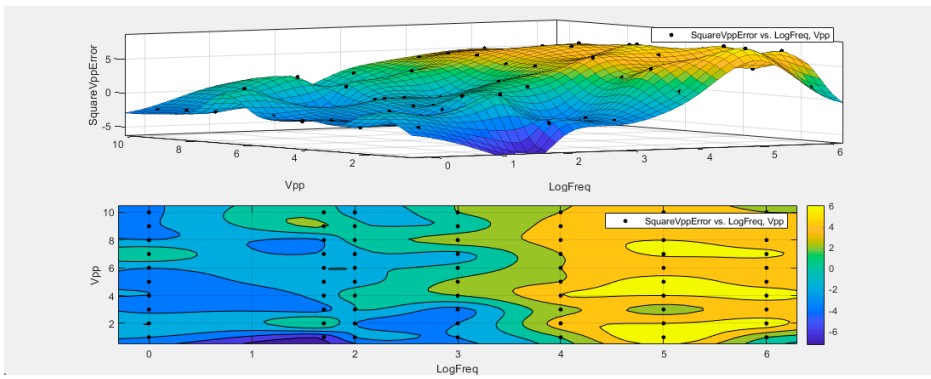

**Figure 14.** Square wave loading amplitude error.

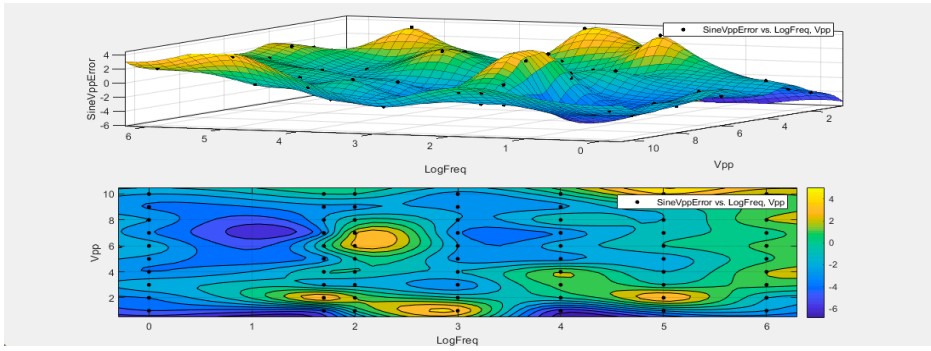

**Figure 15.** Sine wave loading amplitude error.

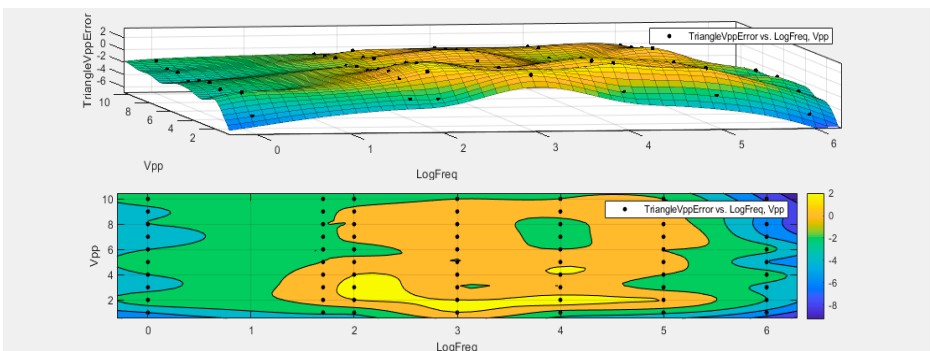

**Figure 16.** Triangular wave loading amplitude error.

### 5.2. Motor Transient Parameter Test

According to the system requirements, the main concerns of the motor transient test include the following: the dynamic test of the motor torque and the dynamic input test of the motor speed signal. Because the feedback signal of the speed torque sensor used in this research is the frequency pulse, in which the speed input frequency range is 0–6 KHz, the torque input frequency range is 5 KHz–10 KHz, and the maximum data update rate is 1 ms. The measured system can measure a frequency ranging from 1 Hz to 1 MHz, and the fastest dynamic refresh rate is 1 ms. Figure 17 shows the speed−step response curve, in which the abscissa is time and the ordinate is motor speed. It can be seen from the curve that the system responds within a millisecond level and reaches a stable state after 1 s. The time to reach stability is also related to the lag effect of the servo motor. The test results show that the system has a fast response speed and preliminarily meets the expected design requirements. Figure 18 shows frequency measurement and analog measurement, which would be used to measure the speed and torque signal of the motor.

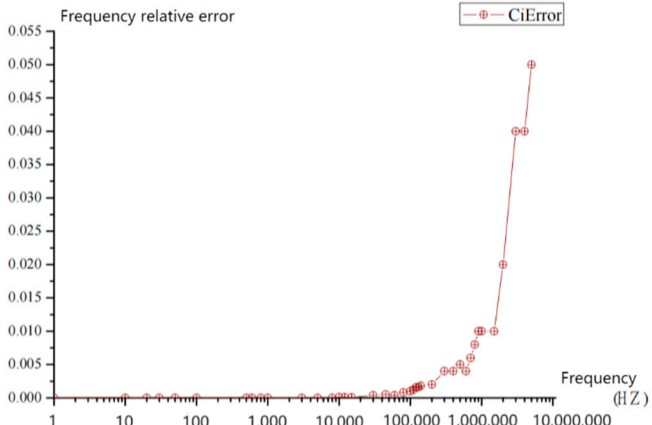

**Figure 17.** Speed and torque frequency accuracy.

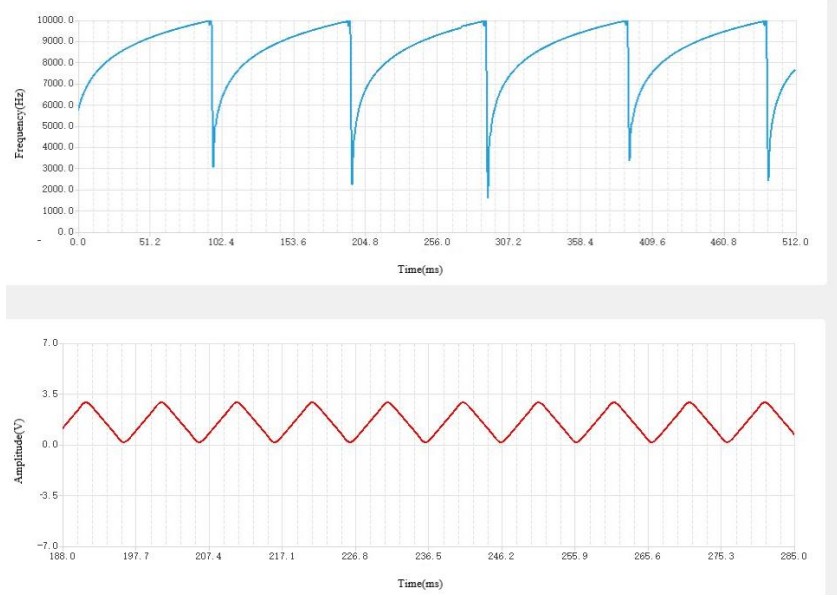

**Figure 18.** The host computer displays the speed frequency and AD sampling waveform in real time.

## 6. Conclusions

Considering the problem that the traditional motor test system cannot directly test the transient parameters of the motor, the requirements of loading a dynamic arbitrary load

when the motor is loaded, as well as the high cost, this paper realizes the main control unit of the motor test system with STM32+FPGA as the core, and FSMC bus communication and the data ping-pong processing algorithm are used to control the dual core and transmit the data stream continuously. The feasibility of the system is verified through its physical design and production. The results show that the project has the following characteristics:

1. It can be embedded into the traditional dynamometer system to upgrade the motor transient test and dynamic loading performance of the original equipment;
2. The overall function of the system can be tailored, and the dynamometer control core based on modular design can independently realize multi-channel or single channel transient test and dynamic loading functions;
3. The uninterrupted transmission of the MS level data stream is realized, and the uninterrupted transmission of the data stream based on the data ping-pong operation algorithm ensures the real-time upload and dynamic refreshing of data.

The core of the dynamometer control system based on STM32+FPGA constructed in this way effectively improves the dynamic free loading of the dynamometer system and the measurement of motor transient parameters. It can be further customized and updated according to customer needs. It can provide a valuable experience for the development of dynamometer technology, especially in the measurement of motor dynamic parameters.

**Author Contributions:** Conceptualization, C.Z.; methodology, C.Z. and B.W.; software, C.Z.; validation, C.Z. and B.W.; formal analysis, C.Z.; investigation, B.W.; resources, C.Z.; data curation, L.B. and B.Z.; writing—original draft preparation, C.Z. and B.W.; visualization, J.Q.; supervision, Y.C.; project administration, C.Z.; funding acquisition, C.Z. All authors have read and agreed to the published version of the manuscript.

**Funding:** This research was funded by The National Natural Science Foundation of China with grant number 62001149 and The Key R&D Program of Zhejiang Province with grant number 2021C01069.

**Conflicts of Interest:** The authors declare no conflict of interest.

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
