# Peer review of "Motor Dynamic Loading and Comprehensive Test System Based on FPGA and MCU"

_electronics, doi:10.3390/electronics11091317_

Round 1

Reviewer 1 Report

Really, I didn't understand why the authors used two systems of ordering and acquisition. you can use only one, FPGA for example. and you can take advantages of the parallelism of the hardware architecture of the FPGA as well as its internal microprocessor (at the FPGA).

The authors should consider adding several recent (last 18 months) references, relevant to this paper, from the Electronics journal. This would help to demonstrate better the link of the paper with this journal and would help its readership to make relevant topical connections. Also, this article requires a comparative study with other works on the same subject.

Reviewer 2 Report

This paper presents a motor test system based on MCU and FPGA. While the system is correctly operate through experiments, I cannot understand its benefits in terns of performance. The comments for this paper is as follows.

  1. Are there any other related works? If so, performance comparison with them are needed.
  2. I cannot understand where the proposed system is used in Figure 1. Please clarify where the proposed system is embedded.
  3. In this paper MCU chip and FPGA are combined. Recently, a CPU-embedded FPGA such as Zynq is easily available. In addition, software-CPU such as Microblaze and NIOSII are also available. How about use such hardware platform for implementing the proposed system.

Round 2

Reviewer 1 Report

ok

Reviewer 2 Report

Thank you for your reply. I can understand the research background and the novelty of this paper. Please highlight modified contents. If no modification, I think minor revision is required so that readers easy to understand your work.